# Assembly of short amphiphilic peptoids into nanohelices with controllable supramolecular chirality

Renyu Zheng[1,2], Mingfei Zhao[3], Jingshan S. Du [2], Tarunya Rao Sudarshan[4], Yicheng Zhou [2], Anant K. Paravastu [4,5], James J. De Yoreo [2,6], Andrew L. Ferguson [3] & Chun-Long Chen [1,2]✉

A long-standing challenge in bioinspired materials is to design and synthesize synthetic materials that mimic the sophisticated structures and functions of natural biomaterials, such as helical protein assemblies that are important in biological systems. Herein, we report the formation of a series of nanohelices from a type of well-developed protein-mimetics called peptoids. We demonstrate that nanohelix structures and supramolecular chirality can be well-controlled through the side-chain chemistry. Specifically, the ionic effects on peptoids from varying the polar side-chain groups result in the formation of either single helical fiber or hierarchically stacked helical bundles. We also demonstrate that the supramolecular chirality of assembled peptoid helices can be controlled by modifying assembling peptoids with a single chiral amino acid side chain. Computational simulations and theoretical modeling predict that minimizing exposure of hydrophobic domains within a twisted helical form presents the most thermodynamically favorable packing of these amphiphilic peptoids and suggests a key role for both polar and hydrophobic domains on nanohelix formation. Our findings establish a platform to design and synthesize chiral functional materials using sequence-defined synthetic polymers.

Chirality is essential in defining the structures and biological processes across length scales. On the macroscale, nautiluses develop spiral shells through biomineralization[1,2]. Collagens that makeup 25% to 35% of the whole human body protein consist of nanoscale helical peptide chains known as triple helices[3,4]. On the molecular scale, the stereo-selectivity of biochemical reactions requires small molecules with specific chirality to proceed and take effect[5–8]. Indeed, such high-level helical structures are often defined by the molecular chirality of their building blocks. Several chiral building blocks, including peptide amphiphiles, $C_3$ symmetric molecules, and π-conjugated molecules, have been reported to assemble into helical structures with supramolecular chirality[9–14]. These examples highlight the importance of understanding how molecular interactions translate the chirality to the geometry of nanostructures by controlling the packing of their structural units[9,13,14].

In the last decades, tremendous efforts have been invested in designing and understanding helical structures assembled from peptides and proteins due to their biological relevance and various

[1]Department of Chemical Engineering, University of Washington, Seattle, WA 98195, USA. [2]Physical Sciences Division, Pacific Northwest National Laboratory, Richland, WA 99352, USA. [3]Pritzker School of Molecular Engineering, University of Chicago, Chicago, IL 60637, USA. [4]School of Chemical and Biomolecular Engineering, Georgia Institute of Technology, Atlanta, GA 30332, USA. [5]Parker H. Petit Institute for Bioengineering and Biosciences, Georgia Institute of Technology, Atlanta, GA 30332, USA. [6]Department of Materials Science, University of Washington, Seattle, WA 98195, USA. ✉e-mail: Chunlong.Chen@pnnl.gov

potential applications[10]. Among those efforts in developing bottom-up approaches for synthesizing helical nanostructures, it is important to have deliberate control over intermolecular interactions[1,13,15–20]. While many peptides and proteins have been used as sequence-defined building blocks for the assembly of helical nanostructures, precise control over their intermolecular interactions remains challenging. In these peptide and protein systems, complex intramolecular and intermolecular interactions lead to rich assembly behaviors and present challenges for rational understanding and prediction of assembly pathways and outcomes[11,21].

Peptoids, or N-substituted glycines which have been developed as a type of peptidomimetics with reduced backbone interactions[22], are sequence-defined polymers structurally similar to peptides, but the side-chain groups reside on the nitrogen instead of the α-carbon. These molecules retain the sequence-programmability and side-chain diversity inherited from peptides but lack backbone hydrogen bond donors and chirality[22,23]. Peptoid-peptoid and peptoid-surface interactions can be tuned by controlling the side-chain chemistry[22,23]. Recently, we and others discovered a series of amphiphilic peptoids that can assemble into diverse nanostructures[11,22–26], including membrane-mimetic nanosheets[27,28] and nanotubes[8,29] which share a similar molecular packing of amphiphilic peptoids akin to the lipid bilayer but are highly crystalline and robust. Despite the recent progress made in the design and synthesis of peptoid-based functional nanomaterials with various nanostructures[11,22,23,30], chiral nanostructures have been rarely reported[31]. While recent computational studies suggest that the formation of membrane-mimetic nanosheets might involve the formation of helical nanostructures as intermediates[32], it remains unclear how a highly ordered, bilayer-like packing of achiral amphiphilic peptoids could be twisted into chiral helices and how to experimentally control the twist and handedness of such assemblies. Murnen et al. reported the assembly of a long achiral amphiphilic peptoid, Npe15Nce15, into homochiral superhelices in 2010[31], but the exact mechanism of these superhelices formation remains unclear due to the experimental challenges in resolving transient intermediates and computational challenges in simulating large biomolecular systems. Thus, further studies to expose molecular understanding of the driving forces for helical structure formation from short achiral peptoids can help advance the predictive synthesis of chiral functional nanomaterials and guide the engineering of peptidomimetic chiral materials with desirable properties for applications including sensing, catalysis, electronics, and photonics.

Herein, we report the design, synthesis, and characterizations of a series of helical nanostructures assembled from short peptoid sequences. By having a short tetrameric hydrophobic side-chain domain and a single polar group to construct the lipid-like amphiphilic peptoids, our results show that the twisted helical structure presents a thermodynamically stable packing of these peptoids in the aqueous environment for their assembly into hierarchical materials. We further find that a slight change of the assembly solution pH condition, or the chemistry of polar group can significantly impact the peptoid assembly outcome, which enables a precise control over the nanohelix handedness by varying the chirality of single polar group. Molecular dynamics (MD) simulations are performed to predict and resolve helical structure formation in atomistic resolution, showing the twisted bilayer-like packing of these amphiphilic peptoids into helical structures shields the hydrophobic side chains from the aqueous solvent, and suggest that a balance of the hydrophobic and hydrophilic interactions is the key for the assembly of these peptoids into helical structures. A simple phenomenological model for the ribbon geometry provides a molecular rationalization for the observed increase in ribbon width upon moving from neutral to acidic pH as a relative change in the balance of peptoid-solvent hydrophobic interfacial interactions and peptoid-peptoid dispersion interactions.

## Results

### Design of short amphiphilic peptoids

Similar to previously reported lipid-like amphiphilic peptoids that self-assemble into membrane-mimetic nanosheets and nanotubes[29,33], we study diblock-like peptoids possessing only four hydrophobic Npm (Npm = N-phenylmethyl glycine) side chains with one polar group, such as a diglycolic acid (Dig) or an alanine (Ala) group, at the N-terminus (Fig. 1a). Due to the decreased number (n) of hydrophobic Npm side chains (from a typical $n = 6$ to 4), we hypothesized, in contrast to our previously reported sheet- and tube-forming sequences[23,29,33], these short peptoids would give us more flexibility to control the self-assembly outcomes for the synthesis of helical nanostructures by varying the chemistry of polar groups and the assembly solution pH conditions (Fig. 1b). All peptoids were synthesized following a previously reported submonomer synthesis method[33,34]. The detailed synthesis and characterizations of these peptoids are shown in the Supporting Information (Supplementary Figs. 1–6).

### Assembly and characterizations of peptoid nanohelices

For the assembly of peptoids into helical nanostructures (Fig. 1b), specific lyophilized peptoids, such as Npm4Dig, were well-dissolved in water and acetonitrile (v/v = 1:1) to obtain a clear solution (5.0 mM). The obtained clear solution was left undisturbed at 4 °C for an easy and slow evaporation to trigger the self-assembly and crystallization process (see Supplementary Information for details). To gain insight into the structure of the peptoid assemblies, we performed computer simulations (See Supplementary Information, Supplementary Fig. 7) and examined the self-assembled materials using transmission electron microscopy (TEM) and atomic force microscopy (AFM). Negatively stained TEM and annular dark-field scanning TEM (ADF-STEM) (Fig. 1c and Supplementary Fig. 8) revealed the presence of nanohelices with a width of $10.7 \pm 1.4$ nm and nearly microns in length. From these TEM images (Fig. 1c and Supplementary Fig. 8), we determined that peptoid nanohelices adopt both a left-handed direction and a right-handed direction with equal populations of each. AFM images (Fig. 1d, e, and Supplementary Fig. 9) further confirmed the formation of peptoid nanohelices, showing a helix thickness of $5.2 \pm 0.4$ nm. Both TEM and AFM results show that these peptoid helices extend in length over several micrometers. The observation of many overlapping helices (Supplementary Figs. 8 and 9) indicated that they were free-standing in solution and overlapped during sample preparation.

To probe for more structural information of the nanohelix, we performed synchrotron-based X-ray diffraction (XRD) and $^{13}$C solid-state nuclear magnetic resonance spectroscopy (ss-NMR) measurements. XRD results show that these peptoid helices are crystalline (Fig. 1e). The first low $q$ peak corresponds to the thickness of bilayer-like packing of Npm4Dig with a spacing of 2.46 nm. Such bilayer-like packing of amphiphilic peptoids is similar to those we previously observed in peptoid nanosheets[33,35] and nanotubes[29] but with much-decreased thickness due to the change of hydrophobic side-chain number (n). The peak at $q = 1.36$ Å$^{-1}$ shows the spacing of 4.6 Å which corresponds to the ordered alignment of peptoid backbone chains, and the 1.36 nm spacing shows the distance between two peptoid backbones packed inside the peptoid bilayer with Npm groups facing each other. The spacing of 2.9 Å could be the distance between two adjacent residues along the backbone chain direction of a *cis*-conformation peptoid[36,37]. To further address how hydrophobic interactions influence the peptoid assembly morphology, we varied the numbers of Npm groups from four to six and two. AFM results showed that the peptoid Npm6Dig with six Npm groups self-assembled into nanosheets (Supplementary Fig. 10a) due to enhanced hydrophobic interactions. In contrast, peptoid Npm2Dig with only two Npm groups self-assembled into isolated particles

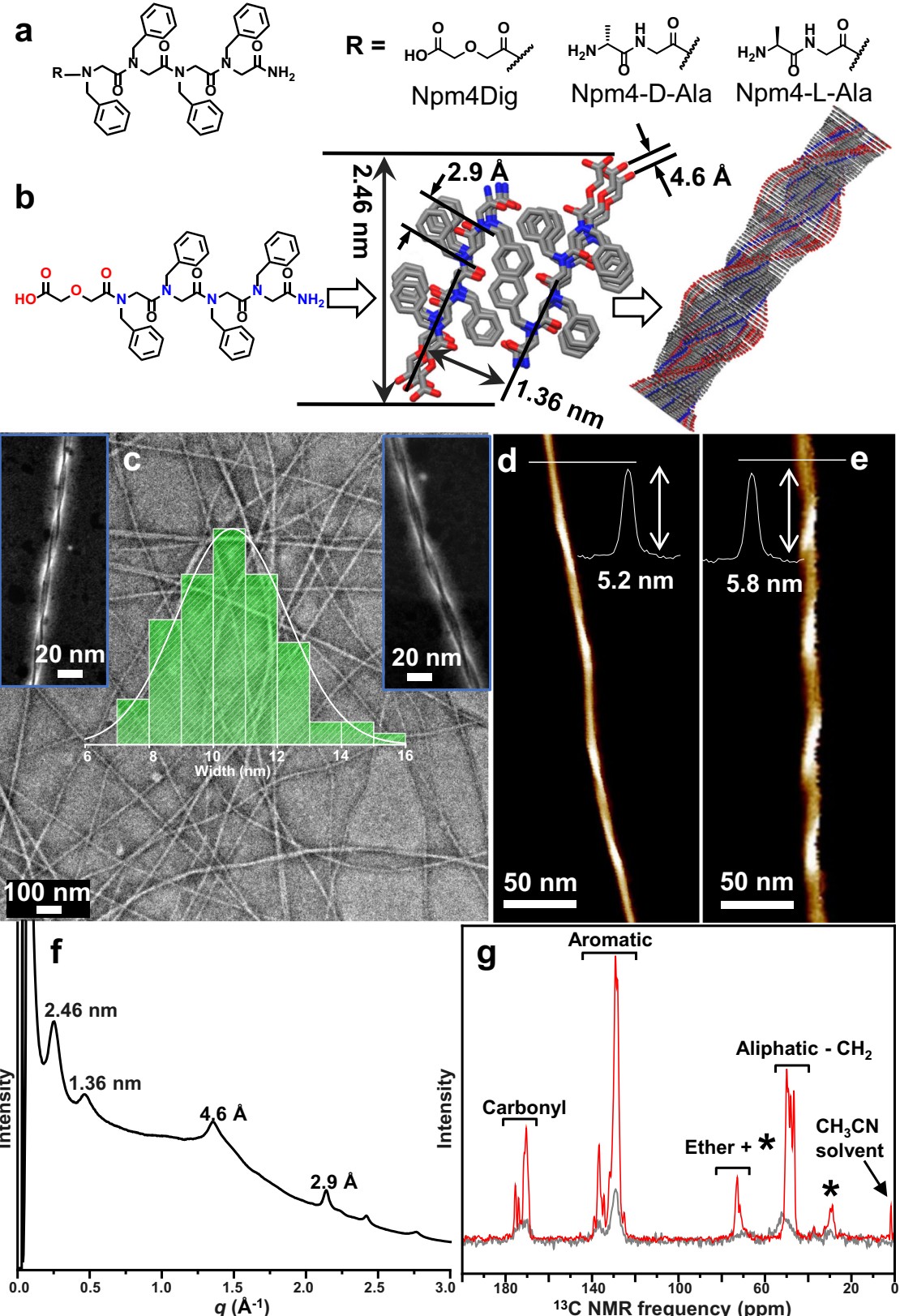

**Fig. 1 | Assembly of amphiphilic peptoids into nanohelices. a** List of three helix-forming peptoids. **b** The scheme showing the assembly of Npm4Dig peptoids into a helical structure; a model of a right-handed helix is listed to show the specific packing of Npm4Dig peptoids. **c** Negative-stained TEM image of Npm4Dig nanohelices. Insets: A high-magnification ADF-STEM image of a left-handed (left), and a right-handed (right) helix, and the width distribution of peptoid nanohelices (middle): 10.7 ± 1.7 nm over 70 measurements. AFM images of one left-handed **d** and one right-handed **e** nanohelix self-assembled from Npm4Dig; Insets: height profiles of one single helix along the section line. **f** X-ray diffraction (XRD) result of the assembled Npm4Dig nanohelices. The values above each peak are the corresponding lattice spacing values. **g** Solid-state $^{13}$C NMR data of Npm4Dig peptoids before (gray color) and after (red color) their assembly into nanohelices; The stars indicate positions of magic-angle spinning sidebands.

(Supplementary Fig. 10b) due to reduced hydrophobic interactions. Such enhanced hydrophobic interactions among Npm6 domains were also revealed in the XRD data of nanosheets assembled from Npm6Nce6 (Supplementary Fig. 11) in which two sharp peaks between d spacings of 2.9 Å and 4.6 Å were observed. Interestingly, similar nanohelices were observed when the assembly of Npm4Dig were performed at different ratios of $CH_3CN$ and $H_2O$ (Supplementary Fig. 12). In contrast to the XRD results of nanosheets assembled from peptoids with six Npm6 groups which have two significant peaks at 4.0 and 3.6 Å (Supplementary Fig. 11), no obvious peaks were observed in this area for self-assembled Npm4Dig nanohelices. Because our previous studies[29,33,37] showed that the peaks occurred in this area are corresponding to the presence of extensive π-stacking among aromatic side-chain groups[38,39], the XRD results suggest that there are no π-stacking interactions among Npm side chains within Npm4Dig nanohelices, which could be due to the less crystallinity feature of these hydrophobic domains within nanohelices compared to those within crystalline nanosheets. Solid-state-NMR results further confirmed that peptoids assembled into ordered structures after nanohelix formation. Specifically, in contrast to the Npm4Dig peptoid samples before the evaporation-induced assembly process, the self-assembled peptoid nanohelix samples demonstrate $^{13}C$ NMR signals with sharper linewidth, indicating assembly into highly ordered structures in aqueous solution[40] (Fig. 1g).

Based on the above TEM, AFM, and XRD results, we arrived a working model for the structure of bilayer-packing of Npm4Dig to form nanohelices (Fig. 1b). While the model of bi-layer packed Npm4Dig is similar to our previously reported models of bi-layer packed lipid-like peptoids that form sheets and tubes[29,33,35], The thickness of this bi-layer peptoid structure (Fig. 1b) is decreased to 2.46 nm due to the reduced number of hydrophobic side chains and the length of peptoids. Furthermore, the twist of Npm side chains could be the reason for the almost no presence of π-stacking observed among these aromatic side chains (Fig. 1f). In contrast to the previously reported superhelices with a diameter of 624 ± 69 nm and the use of peptoid with 30 side chains[31], these nanohelices self-assembled from Npm4Dig offer a system to understand the origin of the helical structure formation due to the much-reduced complexity.

## Molecular dynamic simulation of peptoid nanohelices

To further explore this peptoid helix packing model and to better understand the formation of helical structures, we conducted MD simulations to investigate the development of peptoid nanohelices. The simulation started with a flat, non-helical ribbon with two layers of lipid-like amphiphilic packing in water (Fig. 2a and Supplementary Data 1). The molecular distances within the model are the same as the bi-layer packing of Npm4Dig shown in Fig. 1b, based on the corresponding peaks observed in the XRD spectrum. Initially, simulations were conducted using Npm4Dig peptoids with the Npm4Dig carboxylate group in the deprotonated $-COO^-$ form. The pre-assembled Npm4Dig ribbon with $-COO^-$ groups quickly disintegrate in the simulation, likely due to the electrostatic repulsion. As a result, remaining simulations were conducted with fully protonated -COOH groups. A discussion of pH-dependent assembly is provided below, and full details of the simulation protocol is provided in the Supplementary Information. Our simulations show that the initial flat peptoid ribbons spontaneously adopt a twisted configuration in just 20 ns (Fig. 2b and Supplementary Data 2). Analysis of the twist angle over the course of the simulations reveals the ribbon to adopt a global twist angle of approximately 98°, corresponding to a predicted pitch per full helical turn of 87 nm (Fig. 2c). The spontaneous adoption of a twist suggests that the twisted helical structure is more stable than the flat, non-helical structure and presents a more favorable packing of the Npm4Dig peptoids. Following a previous study on twisted peptide ribbons[41], in which the self-assembly is often governed by a combination of hydrophobic interactions and the propensity of the peptide to form a β-sheet-type hydrogen bonding arrangement, we propose that twisting of the Npm4Dig helices results from shielding the hydrophobic blocks of the peptoids from solvent while maintaining hydrophobic interactions between Npm side chains. As described below, this phenomenological model also explains the observed increase in ribbon width with lowering pH. The simulations predict that the twisted helical structure distributes the hydrophilic Dig groups around the twisted nanoribbons as shown in Fig. 2b. The four independent MD simulations all formed a right-handed helix since they all started from the same initial conditions. However, since Npm4Dig peptoids and water solvent are both achiral, one would expect the right- and left-handed helices to be equally favorable. Indeed, experimentally we

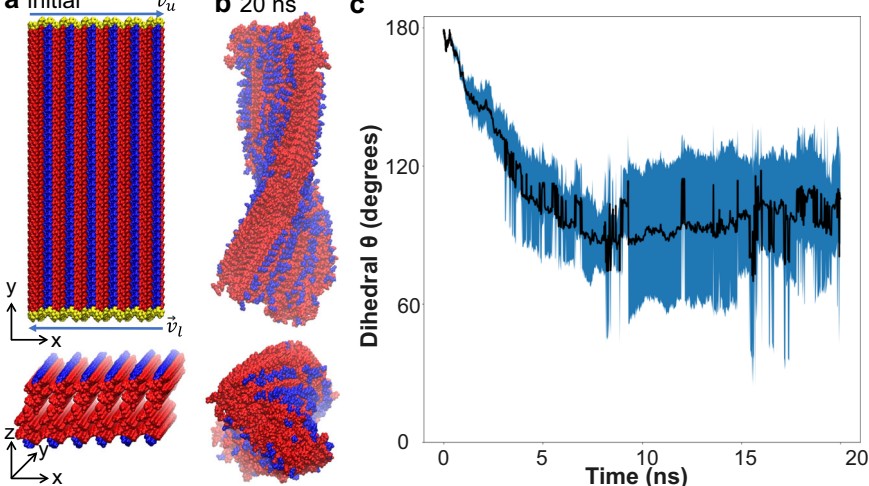

**Fig. 2 | Molecular dynamics (MD) simulations predict the formation of peptoid nanohelices. a, b** MD simulations show the development of a helical structure from the non-helical ribbon structure of bi-layer packed Npm4Dig peptoids. **c** The twist dihedral of the ribbon structure as a function of time. The twist dihedral θ is defined via the relationship $\vec{v}_u \cdot \vec{v}_l = |\vec{v}_u||\vec{v}_l|\cos\theta$, where $\vec{v}_u$ is the best fit vector connecting the center of mass of the peptoids defining the upper edge of the ribbon and $\vec{v}_l$ is the analogous vector on the lower edge. The edge peptoids are colored in yellow shown in (**a**). Error bars indicate standard deviations computed over four independent runs.

observe the formation of left-handed and right-handed helices with equal probability (Supplementary Figs. 8 and 13).

MD simulations suggest that – similar to the formation of peptoid nanosheets and nanotubes[22,29,33,35,42–45] – formation of the Npm4Dig helices is governed by the interplay of the hydrophilic and hydrophobic domains with the aqueous solvent. Thus, we reasoned that a slight change in the hydrophilic domain while keeping the hydrophobic Npm4 domain unchanged could dramatically influence the driving force that leads to the formation of peptoid helices. To test that, we were motivated to vary the assembly solution pH conditions and the chemistry of polar group and demonstrated the experimental synthesis of peptoid nanohelices with controllable thicknesses and chirality.

We first performed the assembly of Npm4Dig in different solution pH because our previous studies showed that the protonation states of carboxylic acid groups could influence the electrostatic interactions and thus impact the packing of the hydrophobic groups during peptoid assembly[46]. At pH = 7, where the carboxylic acid group of Dig is supposed to be deprotonated in the -COO⁻ form, it resulted in increased electrostatic repulsion that drives the formation of individual narrow helices (Fig. 3a). When the solution pH was changed from 7 to 4, the hydrophilic Dig group became partially protonated with a reduced electrostatic repulsion, which makes this hydrophilic group less efficient to cover the hydrophobic domains of Npm4, thus resulting in the formation of nanohelices with multi-layered stacking of twisted ribbons with a much larger widths and heights (Fig. 3b). The quantitative analysis for the heights of multiple helices shows a significant increase of helix height from $5.2 \pm 0.4$ nm to $15.6 \pm 2.4$ nm (Fig. 3d, e) with a 1:1 ratio of left- and right-handedness helices (Supplementary Figs. 13, 14). Interestingly, the experimentally observed helical pitch of $86.4 \pm 6.7$ nm reported at pH = 7 in Fig. 3c is in excellent agreement with the pitch of 87 nm predicted in our MD simulations with all protonated -COOH group. The Npm4Dig carboxylate group exists in a dynamic equilibrium between -COO⁻ and -COOH forms. In lower pH conditions, the equilibrium shifts towards the -COOH form resulting in increased hydrophobicity, and we observe wider nanohelices. In higher pH conditions, the equilibrium shifts towards the -COO⁻ form resulting in increased hydrophilicity that results in narrower nanohelices and, at sufficiently high pH, no ordered self-assembled nanostructures at very high pH conditions. To prove this point, we repeated self-assembly experiments at a high pH condition (pH = 12), which gives amorphous aggregates instead of any ordered self-assembled nanostructures, most likely due to the highly charged polar groups that are sufficient to hide hydrophobic Npm groups in a particle-like aggregation state as a result of the increased electrostatic repulsion (Supplementary Fig. 15). The observation of amorphous aggregates at pH = 12 is in agreement with our MD simulations of peptoids in the -COO⁻ form for which we observed rapid disintegration of a pre-assembled ribbon. A deficiency of our MD simulations is that we model the Npm4Dig carboxylate group in either the -COO⁻ or -COOH form, and so do not directly probe the pH-dependent dynamic equilibrium due to the relatively high computational cost of constant-pH simulations[47]. Taken together, our experimental observations and computational predictions suggest that the greater propensity for aggregation at low pH is due to increased protonation of the Npm4Dig carboxylate group into the -COOH form leading to reduced intermolecular electrostatic repulsion. Conversely, at high pH the deprotonated -COO⁻ form suppresses aggregation into ribbons and instead results in amorphous aggregation. Because the pKa of carboxylic acid group is ~3.5[48], and our results suggest that partial protonation of carboxylate group into the -COOH form occurs at as high as pH 7, we reasoned that the dissociation equilibrium may be substantially modulated due to intramolecular effects within supramolecular stack and exclusion of water solvent within the peptoid helix assemblies.

To gain molecular insight into the observed trend of increasing ribbon width with lowering solution pH, we developed a simple phenomenological thermodynamic model for the geometry of the self-assembled twisted ribbon as an adaptation of that originally developed by Boden and coworkers for twisted β-sheet stacks of peptides[49,50] and subsequently sophisticated by Rüter et al.[41]. The model is based on a balance of interfacial tension and dispersion interactions between peptoid monomers: favorable hydrophobic association of the peptoids is driven by interfacial free energy that promotes the formation of wide twisted ribbons (i.e., nanohelices with large height, cf. Fig. 3b), whereas poorer stacking between peptoid monomers at large ribbon diameters imposes an energetic penalty on the dispersion interactions favors thin twisted ribbons (i.e., nanohelices with small height, cf. Fig. 3b). The critical parameter controlling the thermodynamically preferred ribbon width predicted by the model is a dimensionless parameter $\kappa = \gamma l \delta / \varepsilon$ specifying the relative strengths of the interfacial free energy and peptoid-peptoid dispersion interactions, where $\gamma$ is the interfacial tension, $l$ is a characteristic length of a peptoid monomer, $\delta$ is the characteristic spacing between peptoid layers in the ribbon, and $\varepsilon$ is the characteristic energy scale for van der Waals dispersion interactions between peptoid monomers. Full details of the model are provided in the Supporting Information (Supplementary Fig. 7). As an accompaniment to the experimental data, we present in Fig. 3f the predictions of the model for the relative probabilities of twisted ribbons of various widths $D$ for values of $\kappa = [0.1, 1.0, 10]$. Low pH conditions shift the chemical equilibrium of the Npm4Dig carboxylic acid group from −COO⁻ to −COOH resulting in an increase in molecular hydrophobicity, an elevation of the surface tension, and an increase in κ. Conversely, high pH conditions favor the hydrophilic deprotonated form and result in a reduction of the hydrophobic interfacial tension and a lower value of κ. The trends in the model predictions are in good agreement with the experimental measurements in Fig. 3d, e, wherein lower pH conditions elevate κ and promote the formation of ribbons of greater thickness. This suggests that solution pH acts as a thermodynamic control of the twisted ribbon geometry by modulating the relative strength of the interfacial interactions of the peptoid monomers with the solvent and the dispersion interactions between peptoid monomers.

**Controlling the supramolecular chirality of peptoid nanohelices**
We next investigated the change of polar group chemistry in influencing the formation of peptoid nanohelices. Because recent studies have shown that the change of chirality of amino acids could be used to impact the handedness of self-assembled peptide nanohelices[21,51,52], we thus replaced the achiral Dig group by L- or D-alanine group as the polar domain of the peptoid molecules (See Supporting Information for the synthesis details). Under a similar assembly condition (see Supporting Information for details), both Npm4-D-Ala and Npm4-L-Ala formed gel-like materials. ADF-STEM, AFM, and scanning electron microscopy (SEM) images confirmed that these gel-like materials contained a large amount of peptoid nanohelices with a dominant one single-handedness (Fig. 4, Supplementary Figs. 16–20), which Npm4-D-Ala led to the formation of left-handed nanohelices while L-Ala led to the formation of nanohelices with all right-handedness. Both XRD (Supplementary Fig. 21) and ¹³C ss-NMR (Supplementary Fig. 22) results showed that these obtained left-handed and right-handed helices exhibited similar XRD peaks and NMR signals, suggesting that the supramolecular chirality of peptoid nanohelices might not have direct correlation with the ordering of peptoids within the helical structures. It is worth mentioning that, while peptides containing L-amino acids are typically associated with the formation of left-handed nanohelices[21,51,52], our results showed that L-Ala modified peptoid Npm4-L-Ala surprisingly formed right-handed nanohelices. Similarly, peptoid Npm4-D-Ala formed nanohelices with left-handedness which is opposite to the typical nanohelices

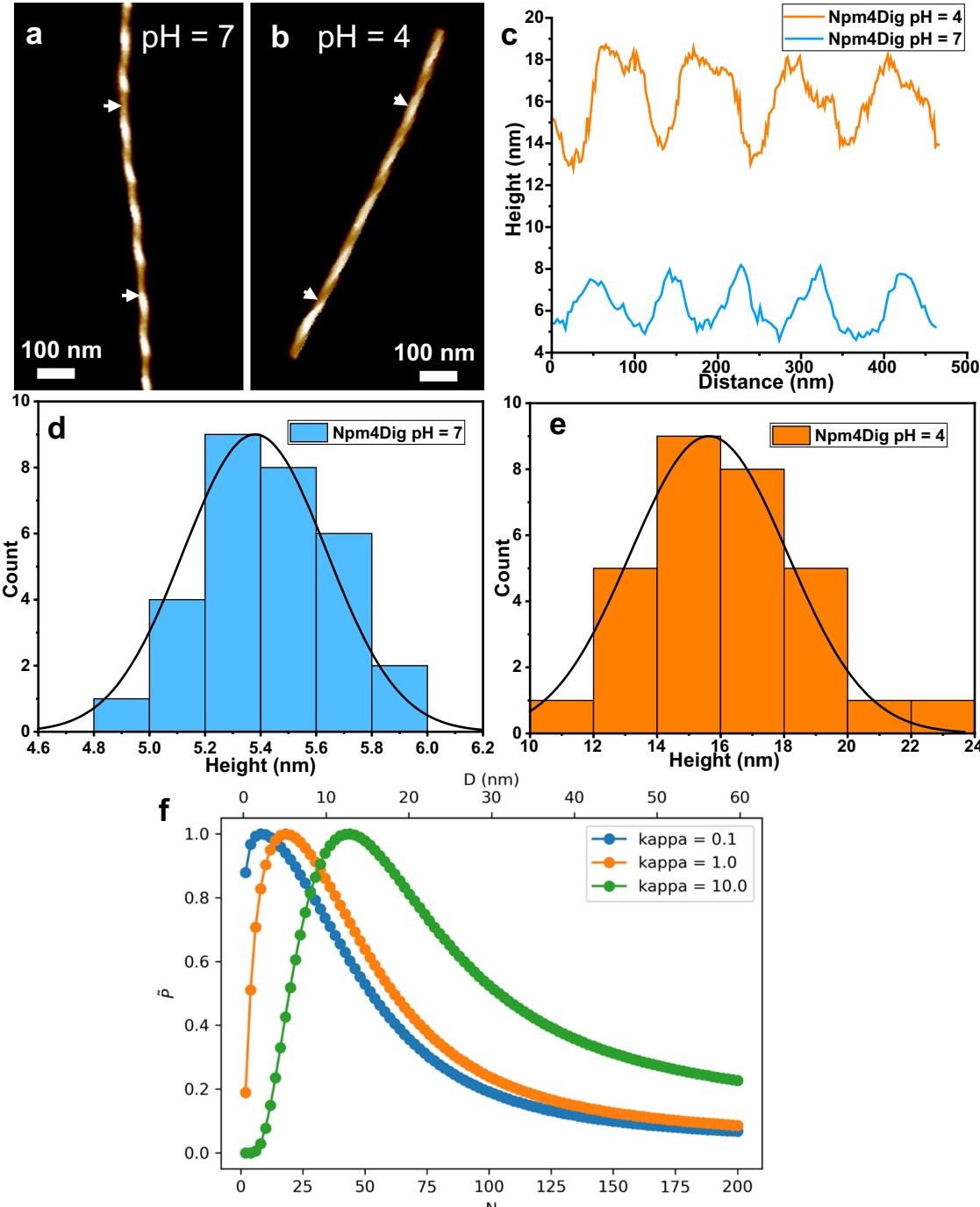

**Fig. 3 | Solution pH-dependent formation of peptoid nanohelices with varied thickness. a**, **b** Npm4Dig nanohelices assembled at pH = 7 (**a**) and pH = 4 (**b**). **c** Height profile and pitch distances along the nanohelix in (**a**) and (**b**), these nanohelices have a regular pitch of 86.4 ± 6.7 nm (pH = 7) and 110.1 ± 20.3 nm (pH = 4), as determined by AFM height images (based on 50 counts). **d**, **e** Height distribution of the peptoid nanohelices assembled at pH = 7 and pH = 4. **f** Predictions of a simple phenomenological model for the relative probabilities $\tilde{P}$ of

twisted ribbons of width $D$ comprising $N$ peptoid monomers across their width as a function of a dimensionless parameter $\kappa = \gamma l \delta / \varepsilon$ specifying the relative strengths of the interfacial free energy and peptoid-peptoid dispersion interactions. Without loss of generality, the curves are scaled such that the maximum relative probability for each curve is unity. Larger values of $\kappa$ shift the distribution towards larger width twisted ribbons.

assembled from peptides composed of D-amino acids[51]. Because these nanohelices assembled from either Npm4-L-Ala or Npm4-D-Ala exhibited a similar framework structure revealed by XRD and ss-NMR results (Supplementary Figs. 21–22), we concluded that the packing of hydrophobic Npm4 domains played a more role in the structural organization than the polar domain. Thus, in these Npm-derived peptoid nanohelix systems, the molecular chirality of Ala amino acid

did not solely control the supramolecular chirality[21] but played important role in the control over the supramolecular chirality of nanohelices together with the ordered structural organization of Npm4 domains. Similar nanohelical structures were observed when the polar side chain was neutral (Supplementary Fig. 23a, b) or a neutral hydroxy group was used as the polar domain (Npm4Noh, Supplementary Fig. 23c, d). In future work, we would like to employ

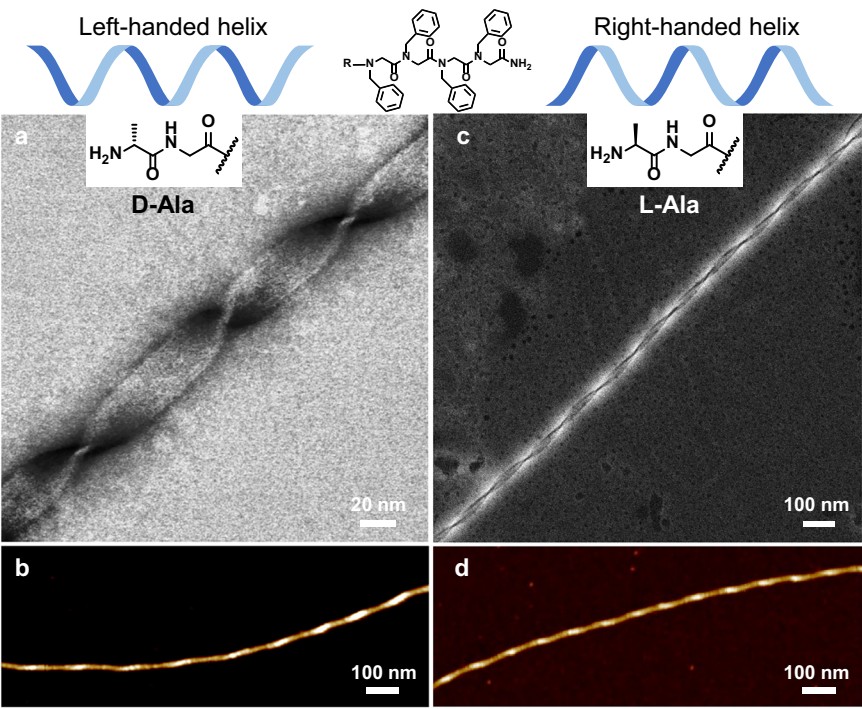

**Fig. 4 | Control over the supramolecular chirality of peptoid nanohelices.**
**a**, **b** Peptoid Npm4-D-Ala with a D-Ala led to the formation of nanohelices with all left-handedness. **c**, **d** Peptoid Npm4-L-Ala with an L-Ala led to the formation of all right-handed nanohelices. Panel **a** negative-stained bright-field STEM image. Panel **c** negative-stained ADF-STEM image. Panels **b** and **d** are AFM images.

enhanced sampling molecular dynamics simulations and free energy calculations to gain molecular-level insight into the relationship between the peptoid side-chain stereochemistry and the emergent supramolecular chirality and endeavor to rationalize the root of the opposite trend to that expected for peptide systems. Our future direction also includes the attachment of the hydrophobic Npm4 domain with other single amino acids, such as hydrophilic aspartic acid with either L- or D-aspartic form, to investigate the influence of single chiral amino acid on the supramolecular chirality of peptoid nanohelices. Therefore, we reason that modifying the hydrophobic domain of short assembling peptoids with single amino acid could provide a platform for controlling the formation of helical nanostructures showing dramatic differences in supramolecular chirality control from those assembled from peptides and proteins.

## Discussion

In conclusion, we reported the assembly of short amphiphilic peptoids into nanohelices with controllable supramolecular chirality. By taking advantage of these short assembling sequences, we have developed a good understanding of the driving forces that led to the formation of chiral helices from achiral peptoids through the combination of experimental mechanistic studies, MD simulations, and theoretical modeling of the helix formation processes. Moreover, we demonstrated that the modification of assembling short peptoids with a single amino acid could lead to the formation of nanohelices with opposite handedness of those assembled from related peptoids, highlighting the significant role of ordered structural packing of peptoid hydrophobic domains in the determination of supramolecular chirality of peptoid assemblies. The ability to precisely control the structural organization of sequence-defined peptoids and their supramolecular chirality of their assemblies opens the door to a system suitable to reveal how molecular interactions between fundamental chemical moieties control the arrangement, packing, and assembly of biomacromolecules and inorganic

particles[1,11,13,15–20,22], generating a wide range of protein-like, functional nanomaterials as nanocarriers for drug delivery[35,44,53], as artificial enzymes for catalytic reactions[8,54], and as scaffolds for biomimetic mineralization[43,55,56].

## Methods

Detailed information on materials and methods is available in the Supplementary Methods. Peptoids were synthesized using a modified solid-phase submonomer synthesis method as described previously[33,54]. The peptoids were synthesized in a 6 mL plastic vial, cleaved from the resin by addition of 95% trifluoroacetic acid in water, and then dissolved in water and acetonitrile (v/v = 1:1) for HPLC purification. Lyophilized and HPLC-grade peptoids were dissolved in the mixture of water and acetonitrile (v/v = 1:1) to make a 5-mM clear solution, which was then transferred to a 4 °C refrigerator for slow evaporation.

## Data availability

All data are available within the Article and Supplementary Files, or available from the corresponding authors on request.

## Code availability

Custom code is available from the corresponding author on request.

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

## Acknowledgements

This work was supported by the US Department of Energy (DOE), Office of Science, Office of Basic Energy Sciences (BES) as part of the Energy Frontier Research Centers program: CSSAS – The Center for the Science of Synthesis Across Scales – under Award Number DE-SC0019288 [FWP 72448 at Pacific Northwest National Laboratory (PNNL)]. The synthesis of peptoid Npm6Nce6 and Npm6Dig nanosheets was supported by DOE-BES biomolecular materials program under an award FWP 65357 at PNNL. XRD work was conducted at the Advanced Light Source (ALS) of Lawrence Berkeley National Laboratory, which was supported by the Office of Science (No. DE-AC02-05CH11231). Electron microscopy was performed on a project award (DOI: 10.46936/cpcy.proj.2022.60575/60008644) from the Environmental Molecular Sciences Laboratory (EMSL) at PNNL. Part of AFM and S/TEM experiments were conducted at the Molecular Analysis Facility, a National Nanotechnology Coordinated Infrastructure (NNCI) site at the University of Washington, which is supported in part by funds from the National Science Foundation (awards NNCI-2025489, NNCI-1542101). This work was completed in part with resources provided by the University of Chicago Research Computing Center. J.S.D. acknowledges a Washington Research Foundation Postdoctoral Fellowship, which supported some S/TEM measurements. PNNL is multi-program national laboratory operated for DOE by Battelle under Contracts No. DE-AC05-76RL01830.

## Author contributions

C.-L.C. conceived and directed the project. R.Z. performed the synthesis and assembly of peptoid materials. R.Z. performed AFM characterization and data analysis. R.Z., J.S.D. and Y.Z. performed S/TEM characterization and data analysis. C.-L.C. did XRD experiments. R.Z. and C.-L.C. analyzed XRD data. M.Z. and A.L.F. conducted the molecular dynamics simulations and the theoretical modeling. T.R.S. and A.P. performed solid-state-NMR experiments and data analysis. J.J.D.Y. discussed S/TEM and AFM results. R.Z., M.Z., A.F., J.S.D., T.R.S., A.P., and C.-L.C. wrote the manuscript. All authors discussed the results and commented on the manuscript.

## Competing interests

A.L.F. is a co-founder and consultant of Evozyne, Inc. and a co-author of US Patent Applications 16/887,710 and 17/642,582, US Provisional Patent Applications 62/853,919, 62/900,420, 63/314,898, 63/479,378, and 63/521,617, and International Patent Applications PCT/US2020/035206 and PCT/US2020/050466. The remaining authors declare no competing interests.
