## [Peer Review File · Nature Communications]

Assembly of short amphiphilic peptoids into nanohelices with controllable supramolecular chiralityEditorial Note: Parts of this Peer Review File have been redacted as indicated to maintain the confidentiality of unpublished data.

REVIEWER COMMENTS

Reviewer #1 (Remarks to the Author):

In this manuscript entitled “Assembly of short amphiphilic peptoids into nanohelices with controllable supramolecular chirality”, Zheng and co-authors developed an interesting study of using short peptoid oligomers to assemble chiral helical materials. These short amphiphilic peptoids enabled them to systematically study their assembly into chiral helical materials. The authors further used molecular dynamics (MD) simulations to predict and resolve the helical structure formation in atomistic resolution. Though the combination of experimental mechanistic studies, MD simulations, and theoretical modeling, the authors have developed a nice understanding of the driving force that leads to the formation of chiral helical structure from sequence-defined achiral peptoids. They further demonstrated the control over the chirality of peptoid helices by using one single chiral side chain to control the formation of nanohelices with specific handedness.

Overall, the research is a very impactful study, and one of the first to comprehensively study how sequence-defined macromolecules assemble into chiral helical materials. Such high-level control over the molecular interactions to precisely control the organization of sequence-defined peptoids into nanohelices with controlled supramolecular chirality could significantly advance the predictive synthesis of chiral functional nanomaterials. The paper is generally well written, and the results are properly discussed, supporting the author hypothesis. Therefore, I would recommend the acceptance of this manuscript after minor revision

- 1) It is not clear the assembly condition for the Npm4-L-Ala or Npm4-D-Ala, will solution pH significantly impact the nanohelix formation there? Will the protonation stage of Npm4-Ala peptoids influence the formation of nanohelices and their supramolecular chirality?
- 2) Besides using Dig group which could be deprotonated, and Ala group which could be protonated as the hydrophilic group, will the helical structure still form if a neutral hydrophilic side chain is used? How significant the packing of hydrophobic N bbbpm4 domains is to determine the formation of nanohelices?
- 3) During the assembly of macromolecules, solvents could play an important role in the packing of macromolecules during the assembly process, have authors investigated their influence of CH₃CN&H₂O ratio or using different solvents to investigate the assembly of these peptoids, will those changes still lead to the formation of nanohelices?
- 4) Although the paper is discussing on the organic chirality, it is better to discuss the peptide or amino acid induced inorganic chirality to broaden the scope of this paper. For instance, the papers Nature Reviews Bioengineering 2023, 1 (2), 88; Nature Reviews Chemistry 2022, 6, 125; Nature 2022, 470, 476; Nature 2018, 556 (7701), 360; Science, 2019, 365(6460), 1475 can be useful for the discussion and citation.

Reviewer #2 (Remarks to the Author):

The manuscript by Zheng et al. reports twisted ribbon “nanohelices” of peptoid (biomimetic sequence-programmable N-substituted oligoglycines) tetramers. This work follows several groundbreaking examples of peptoid nanostructures from the Chun-Long Chen group. The current work is significant as it is the first such (nano)structure reported for peptoids; moreover, the driving forces for assembly are understood in enough detail to achieve molecular control of helical geometry and supramolecular chirality. The conclusions of the study are well supported by atomic force microscopy, spectroscopic and computational studies.

Several aspects of the manuscript should be particularly celebrated. For one, the computational modeling is expertly done. It is especially satisfying that transferrable all-atom molecular simulations (and not coarse-grained models) are able to excellently reproduce the experimentally measured twist dihedral. For another, the model of assembly led to testable hypotheses about how the protonation states of the hydrophilic domains would affect assembly. Also, the phenomenological model used to explain the thermodynamic modulation of the twisted ribbon geometry offers particularly useful insights.

Below are some issues that should be addressed by the authors:

Are the only “solvent conditions” being tuned here pH ? If so, it might be better just to say that directly. “Solvent conditions” is vague.

While there have been many twisted ribbon nanostructures/helices reported in the literature, it would appear that this is one of the few examples of such structures for peptoids. Can the authors be a bit more specific on how these nanohelices structures compare with the earlier nanohelices reported by Murnen et al. (size, composition, dispersion, etc.). Also, such structures have been observed/engineered for peptides (e.g. work from the Olsson group); could the authors elaborate on how the driving forces for assembly differ from that of peptides?

The authors found that L- versus D-alanine substitutions produced the opposite supramolecular chirality as seen for peptides. The authors suggest the subtleties of hydrophobic packing can explain this. Is there anything to be learned from the computational modeling in this case? Even a simple geometric argument? Would alchemical free energy perturbation or other approaches help reveal how such changes affect preferences for left- vs right-handed helices?

In the Figure 1 caption, it is somewhat unclear that the left and right helices in panel d) are both examples of Npm4Dig. Could the authors make this specifically clear?

Some minor editorial *suggestions* are below:

Abstract, line 19: Change “with the extreme importance” to “that are extremely important”

Abstract, line 22: Change “with varying” to “from varying”

Introduction, lines 42-43: Remove “and recognitions”

Introduction, line 48: Change “significant” to “important”

Reviewer #3 (Remarks to the Author):

Chen et al reported ‘Assembly of short amphiphilic peptoids into nanohelices 1 with controllable supramolecular chirality’. The work describes peptoid tetramer with versatile N-terminal substitutions for versatile nanohelix formation and negatively-stained ADF-STEM images are nicely taken.

The concept of similar nanohelix formation has been extensively reported by Zuckermann previously: Mannige, R., Haxton, T., Proulx, C. et al. Peptoid nanosheets exhibit a new secondary-structure motif. *Nature* 526, 415–420 (2015). <https://doi.org/10.1038/nature15363>; Hierarchical supramolecular assembly of a single peptoid polymer into a planar nanobrush with two distinct molecular packing motifs, <https://doi.org/10.1073/pnas.2011816117>.

Maayan et al showed altering the substitution in peptoid can form versatile scaffolds for metallopeptoids supported by x-ray and TEM analysis, From Distinct Metallopeptoids to Self-Assembled Supramolecular Architectures, <https://doi.org/10.1002/chem.202003612>.

Thus, the lack of novelty is the major drawback of the present work.

Moreover, there are several technical drawbacks:

1. Without a definite x-ray structure the claim of peptoid orientation can't be confirmed just using MD simulation.
2. why only peptoid tetramer was used, a justification is missing. Definite control experiments without using Npm are lacking.
3. regarding the experiment with L-Ala and D-Ala is too primary. A few more amino acids especially with Phe and Lys in both L/D forms are required to comment on the reasons for the altering helices.
4. ‘ Because our previous studies showed that the peaks that occurred in this area are corresponding to the presence of extensive π -stacking among aromatic side chain groups, the XRD results suggest that there are no π -stacking interactions among Npm side chains within Npm4Dig nanohelices.’ (Line 146-148) the reason why no π -stacking interactions have been observed is required.
5. ‘we thus replaced the achiral Dig group by L- or D-alanine group as the hydrophilic terminal of the peptoid molecules’ (Line 291-292), is it a typo that ala would be hydrophilic terminal as Ala is known to be hydrophobic?
6. ‘Supramolecular chirality’ is not supported with enough experiments, especially concentration-dependent CD is highly recommended.
7. ESI-MS ranges can be shown from 200-900 Da range for Fig S3. For Fig. S1, can you explain what is the peak for 459?
8. Formation of peptoid scaffold computationally could not be supported unless the lowest energy conformational search is performed.

9. As a force field why CHARMM 22 is used could be explained in SI, comparing at least one geometry with CHARMM36 or NAMD.

10. The referencing is not at all updated for the peptoids, crucial contributions in the field by others have not been acknowledged.

11. I wonder where and how this tetramer peptoid can be used for 'generating a wide range of protein-like, functional nanomaterials as drug delivery agents, artificial enzymes, and tissue engineering scaffolds'? How can this tetramer contribute to drug delivery? Or tissue engineering? A few possible references or any primary experiment for the same would be nice to see as an application.

The conclusion part is poorly written, especially the last portion is misleading without proper experimental support.

Overall, lack of novelty, several major technical experiments are missing, no application of the helices has been shown and misleading conclusions do not warrant publication of this manuscript in this journal.

Reviewer #1

“Overall, the research is a very impactful study, and one of the first to comprehensively study how sequence-defined macromolecules assemble into chiral helical materials. Such high-level control over the molecular interactions to precisely control the organization of sequence-defined peptoids into nanohelices with controlled supramolecular chirality could significantly advance the predictive synthesis of chiral functional nanomaterials. The paper is generally well written, and the results are properly discussed, supporting the author hypothesis. Therefore, I would recommend the acceptance of this manuscript after minor revision.”

We thank the reviewer for noting the quality of our work. We have now addressed all comments in full.

1) *“It is not clear the assembly condition for the Npm4-L-Ala or Npm4-D-Ala, will solution pH significantly impact the nanohelix formation there? Will the protonation stage of Npm4-Ala peptoids influence the formation of nanohelices and their supramolecular chirality?”*

Response: We thank the reviewer for pointing out this confusion. For the assembly experiments in Fig.4, we did the self-assembly of both Npm4-L-Ala and Npm4-D-Ala sequences at pH=4. Because under this condition, the -NH₂ amino groups were protonated, thus we conclude that the protonation stage of Npm4-Ala won't disrupt the formation of nanohelices and their supramolecular chirality. To address the question about the potential pH impact for peptoid nanohelix assembly, we further did the self-assembly for Npm4-L-Ala at pH=10, which the polar amino group was deprotonated. As shown in the Figure below, Npm4-L-Ala formed a similar nanohelix structure with right handedness.

Fig. R1 Negatively stained TEM images of Npm4-L-Ala nanohelices self-assembled at pH=10.

To address this comment, we now added the pH conditions in the updated Supplementary Figure Captions and added this above Figure as **Supplementary Figure 23a and b**.

2) "Besides using Dig group which could be deprotonated, and Ala group which could be protonated as the hydrophilic group, will the helical structure still form if a neutral hydrophilic side chain is used? How significant the packing of hydrophobic Npm4 domains is to determine the formation of nanohelices?"

Response: We thank this reviewer for these great questions.

As we showed in the newly added data (Supplementary Figure 23a and b), similar helices formed when the polar domain is not protonated, suggesting that the helical structure still forms when the polar side chain is neutral. To address this reviewer's comment, we also synthesized peptoid, Npm4Noh, with a neutral hydroxy group as the polar domain. As shown in Supplementary Figure 23c and d, this peptoid Npm4Noh also formed a similar helical structure.

Supplementary Figure 23 | Peptoid nanohelices self-assembled from peptoids with different polar domains. a,b Negatively stained TEM images of Npm4-L-Ala nanohelices self-assembled at pH = 10. **c**) Negatively stained TEM image of Npm4Noh nanohelices self-assembled at pH = 4. **d**) AFM image of Npm4NOH nanohelices self-assembled at pH = 4. The insert in each figure is the chemical structure of the corresponding peptoid.

To address the second question, we further synthesized peptoid Npm6Dig and Npm2Dig by keeping the same Dig polar domain while changing the hydrophobic domain from Npm4 to Npm6 or Npm2. As shown in the newly added Figure below (Supplementary Figure 10), peptoid Npm6Dig formed nanosheets (Supplementary Figure 10a) with no helical structure observed. When the number of hydrophobic Npm groups was increased from 4 to 6, the hydrophobic domains of assembling peptoids formed enhanced hydrophobic interactions which promoted the stabilization of nanosheet structure. In contrast, Npm2Dig formed isolated particles due to the significantly weakened hydrophobic interactions (Supplementary Figure 10b).

Supplementary Figure 10 | AFM images of peptoid assemblies formed from assembling peptoids with same polar domain but different hydrophobic domain with varied numbers of Npm groups. a) AFM image of nanosheets self-assembled from Npm6Dig. b) AFM image of isolated particles self-assembled from Npm2Dig.

Such enhanced hydrophobic interactions were also revealed in the XRD data of nanosheets assembled from Npm6Nce6 in the Supplementary Figure 11, two sharp peaks between d spacings of 0.29 nm and 0.46 nm were observed.

Supplementary Figure 11 | XRD data of nanohelices assembled from Npm4Dig and nanosheets assembled from Npm6Nce6.

To address this comment, we made the following changes to the updated manuscript:

“... a cis-conformation peptoid. To further address how hydrophobic interactions influence the peptoid assembly morphology, we varied the numbers of Npm groups from four to six and two. AFM results showed that the peptoid Npm6Dig with six Npm groups self-assembled into nanosheets (Supplementary Fig. 10a) due to enhanced hydrophobic interactions. In contrast, peptoid Npm2Dig with only two Npm groups self-assembled into isolated particles (Supplementary Fig. 10b) due to reduced hydrophobic interactions. Such enhanced hydrophobic interactions among Npm6 domains were also revealed in the XRD data of nanosheets assembled from Npm6Nce6 (Supplementary Fig. 11) in which two sharp peaks between d spacings of 2.9 Å and 4.6 Å were observed...”

3) “During the assembly of macromolecules, solvents could play an important role in the packing of macromolecules during the assembly process, have authors investigated their influence of CH₃CN&H₂O ratio or using different solvents to investigate the assembly of these peptoids, will those changes still lead to the formation of nanohelices?”

Response: We appreciate this reviewer for pointing out the impact of solvents for the assembly of peptoids. To address this comment, we have tested the assembly of Npm4Dig at different ratios of CH₃CN and H₂O. In addition to the self-assembly of Npm4Dig in 1:1 CH₃CN&H₂O, we also performed the self-assembly of Npm4Dig in 1:2 and 2:1 CH₃CN&H₂O. As shown in the newly added data (Supplementary Figure 12), similar nanohelices of peptoid Npm4Dig were observed, suggesting that CH₃CN&H₂O ratio did not obviously impact the formation of peptoid nanohelices.

Supplementary Figure 12 | Peptoid nanohelices self-assembled from peptoid Npm4Dig with different ratios of CH₃CN and H₂O. a) Negatively stained TEM image of nanohelices self-assembled from Npm4Dig in 1:2 CH₃CN&H₂O, (b) Negatively stained TEM image of nanohelices self-assembled from Npm4Dig in 2:1 CH₃CN&H₂O.

In our ongoing study of solvent-controlled assembly of Npm4-L-Ala, our preliminary results showed that Npm4-L-Ala could form nanosheets [REDACTED], nanotubes [REDACTED] besides nanohelices. [REDACTED], we hypothesized that the solvents with [REDACTED] could offer a great opportunity for tuning the morphologies of these peptoid assemblies and offer a better understanding of the transformations among peptoid helices, tubes and nanosheets. Such systematic studies are beyond the scope of this manuscript and will be published in the follow-up manuscript.

[REDACTED]

To address this comment, we added a new **supplementary Figure 12** and added this sentence in the revised manuscript:

“Interestingly, similar nanohelices were observed when the assembly of Npm4Dig were performed at different ratios of CH₃CN and H₂O (Supplementary Fig. 12).”

5) *“Although the paper is discussing on the organic chirality, it is better to discuss the peptide or amino acid induced inorganic chirality to broaden the scope of this paper. For instance, the papers Nature Reviews Bioengineering 2023, 1 (2), 88; Nature Reviews Chemistry 2022, 6, 125; Nature 2022, 470, 476; Nature 2018, 556 (7701), 360; Science, 2019, 365(6460), 1475 can be useful for the discussion and citation.”*

Response: We thank the reviewer for pointing out these interesting papers. To address this comment, we made the following changes in the updated introduction by citing these papers:

“Among those efforts in developing bottom-up approaches for synthesizing helical nanostructures, it is important to have deliberate control over intermolecular interactions.”^{1,13,15-20}

Besides, we believe that the assembly of sequence-defined peptoids into nanohelices with controlled supramolecular chirality could serve as a platform of developing organic templates to induce the formation of inorganic chirality. For that, we also made some changes in the last paragraph of “Discussion” and cited these papers there:

chirality of their assemblies opens the door to a unique system suitable to reveal how molecular interactions between fundamental chemical moieties control the arrangement, packing, and folding assembly of biomacromolecules and inorganic particles,^{1,11,13,15-20,22} generating a wide range of protein-like, functional nanomaterials as nanocarriers for drug delivery agents,^{35,44,53} as artificial enzymes for catalytic reactions,^{8,54} and as tissue engineering scaffolds for biomimetic mineralization.^{43,55,56}

Reviewer #2

“The manuscript by Zheng et al. reports twisted ribbon “nanohelices” of peptoid (biomimetic sequence-programmable N-substituted oligoglycines) tetramers. This work follows several groundbreaking examples of peptoid nanostructures from the Chun-Long Chen group. The current work is significant as it is the first such (nano)structure reported for peptoids; moreover, the driving forces for assembly are understood in enough detail to achieve molecular control of helical geometry and supramolecular chirality. The conclusions of the study are well supported by atomic force microscopy, spectroscopic and computational studies.

Several aspects of the manuscript should be particularly celebrated. For one, the computational modeling is expertly done. It is especially satisfying that transferrable all-atom molecular simulations (and not coarse-grained models) are able to excellently reproduce the experimentally measured twist dihedral. For another, the model of assembly led to testable hypotheses about how the protonation states of the hydrophilic domains would affect assembly. Also, the phenomenological model used to explain the thermodynamic modulation of the twisted ribbon geometry offers particularly useful insights.”

We appreciate this reviewer for the very positive feedback and comment. We have now addressed all comments in full.

1) “Are the only “solvent conditions” being tuned here pH? If so, it might be better just to say that directly. “Solvent conditions” is vague.”

Response: We thank the reviewer for pointing out this. To address this comment, we have now changed “the assembly solution conditions” to

“the assembly solution pH conditions”

2) “While there have been many twisted ribbon nanostructures/helices reported in the literature, it would appear that this is one of the few examples of such structures for peptoids. Can the authors be a bit more specific on how these nanohelices structures compare with the earlier nanohelices reported by Murnen et al. (size, composition, dispersion, etc.). Also, such structures have been observed/engineered for peptides (e.g. work from the Olsson group); could the authors elaborate on how the driving forces for assembly differ from that of peptides?”

Response: Compared with the earlier peptoid nanohelix structure reported by Murnen et al. (*J. Am. Chem. Soc.* 2010, **132**, 16112-16119; original reference 22 in the manuscript), our peptoid nanohelices have two main differences: First, the heights and widths of our nanohelices are at the range of 10-30 nm, compared with the large superhelices with a diameter of 624 ± 69 nm in Murnen et al.'s work. Second, we use the short-sequence peptoid tetramer, while Murnen et al. used long-sequence peptoids containing 30 side groups. In Murnen et al.'s work, the origin of this helix remains a mystery due to the experimental challenges in resolving the structures of transient intermediates and computational challenges in simulating their assembly process as a result of the large size of 30-mer peptoid. Therefore, our peptoid nanohelix system with a short assembling peptoid make it possible to have a better understanding of the structural information of transient intermediates and construct the helical structure by using molecular dynamics (MD) simulation, thus revealing the driving force of the helix formation.

To address this comment, we made following changes in the revised manuscript:

of π -stacking observed among these aromatic side chains (Fig. 1f). In contrast to the previously reported superhelices with a diameter of 624 ± 69 nm and the use of peptoid with 30 side chains,³¹ these nanohelices self-assembled from Npm4Dig offers a unique system to understand the origin of the helical structure formation due to the much reduced complexity.

Molecular dynamic simulation of peptoid nanohelices

In peptide nanohelices, inter- and intramolecular hydrogen bonds are crucial for the nanohelix formation. More specifically, the self-assembly is thought to be governed by a combination of hydrophobic interactions and the propensity of the peptide to form a β -sheet-type hydrogen bonding arrangement. In addition, the twisting direction of the peptide helices can be influenced by the dihedral angle developed from the β -sheet packing. Therefore, hydrogen bonds become indispensable when analyzing the formation mechanism of peptide helices. However, our peptoid assembly system has no backbone hydrogen bond donors, and the self-assembly of peptoids is governed by hydrophobic and electrostatic interactions. It allows us to use a simple but effective model to describe the self-assembly behavior, as we have done in this manuscript. Besides, we believe excluding hydrogen bonds makes our conclusion applicable to other synthetic polymer systems.

To address this comment, we made this following change:

“...Following a previous study on twisted peptide ribbons⁴¹, which the self-assembly is often governed by a combination of hydrophobic interactions and the propensity of the peptide to form a β -sheet-type hydrogen bonding arrangement, we propose that twisting of the Npm4Dig helices results from...”

3) *“The authors found that L- versus D-alanine substitutions produced the opposite supramolecular chirality as seen for peptides. The authors suggest the subtleties of hydrophobic packing can explain this. Is there anything to be learned from the computational modeling in this case? Even a simple geometric*

argument? Would alchemical free energy perturbation or other approaches help reveal how such changes affect preferences for left- vs right-handed helices?"

Response: The supramolecular chirality of peptoid nanostructures induced from L/D-alanine substitutions is indeed a very interesting observation that could be probed further by molecular simulations. We have not conducted any molecular simulations of the L/D-alanine substitutions and, before doing so, we would likely need to validate that the Weiser and Santiso peptoid force field employed in this work is capable of satisfactorily modeling L and D chirality side chains (We suspect that it is but will have to test this). We have previously employed computational modeling to identify how amino acid stereochemistry affects the supramolecular chirality of self-assembled pi-conjugated peptides (<https://dx.doi.org/10.1021/acs.langmuir.0c00961>) but found that the effects were so subtle. On the other side, such computational modeling required extensive enhanced sampling calculations that were so expensive that we had to be obliged to analyze the system behavior at the level of a dimer as a proxy for the full multi-body assembly. It is our anticipation that similar costly calculations would be required to garner molecular level insight into the supramolecular chirality and, ideally, as the reviewer suggests, be complemented with free energy calculations to quantify the relative thermodynamic preference for the two supramolecular chiral forms. Due to the anticipated very high computational cost for these calculations, we respectfully prefer to defer these computations to a future simulation-centric work, but fully agree with the reviewer that these may be illuminating and useful studies that we should illuminate for the reader. To this end, we added the following sentences in the revised manuscript:

chirality of nanohelices together with the ordered structural organization of Npm4 domains.

Similar nanohelical structures were observed when the polar side chain was neutral

(Supplementary Fig. 23a, b) or a neutral hydroxy group was used as the polar domain (Npm4Noh,

Supplementary Fig. 23c, d). In future work, we would like to employ enhanced sampling molecular

dynamics simulations and free energy calculations to gain molecular-level insight into the

relationship between the peptoid side chain stereochemistry and the emergent supramolecular

chirality and endeavor to rationalize the root of the opposite trend to that expected for peptide

systems. Therefore, we propose that the modification of short assembling peptoid sequences could

4) "In the Figure 1 caption, it is somewhat unclear that the left and right helices in panel d) are both examples of Npm4Dig. Could the authors make this specifically clear?"

Response: Thanks for pointing out this, we now fixed that and made this following change:

"...AFM images of one left-handed (d) and one right-handed (e) nanohelix self-assembled from Npm4Dig; Insets: height profiles of one single helix along the section line."

5) Some minor editorial **suggestions** are below:

Abstract, line 19: Change “with the extreme importance” to “that are extremely important”

Abstract, line 22: Change “with varying” to “from varying”

Introduction, lines 42-43: Remove “and recognitions”

Introduction, line 48: Change “significant” to “important”

Response: Thanks for pointing out these changes, all were corrected accordingly in the revised manuscript.

Reviewer #3

1) “Chen et al reported ‘Assembly of short amphiphilic peptoids into nanohelices 1 with controllable supramolecular chirality’. The work describes peptoid tetramer with versatile N-terminal substitutions for versatile nanohelix formation and negatively-stained ADF-STEM images are nicely taken.

The concept of similar nanohelix formation has been extensively reported by Zuckermann previously: Mannige, R., Haxton, T., Proulx, C. et al. *Nature* 526, 415–420 (2015). <https://doi.org/10.1038/nature15363>; Hierarchical supramolecular assembly of a single peptoid polymer into a planar nanobrush with two distinct molecular packing motifs, <https://doi.org/10.1073/pnas.2011816117>.

Maayan et al showed altering the substitution in peptoid can form versatile scaffolds for metallopeptoids supported by x-ray and TEM analysis, *From Distinct Metallopeptoids to Self-Assembled Supramolecular Architectures*, <https://doi.org/10.1002/chem.202003612>.

Thus, the lack of novelty is the major drawback of the present work.”

Response: We thank the reviewer for highlighting these important studies in the field of peptoid assembly research. However, we would like to clarify the concept of “nanohelix” and emphasize the novelty of our manuscript compared to previous papers that this reviewer mentioned. The nanohelix we reported here refers to the supramolecular self-assembly morphology and is parallel to other morphologies, such as nanosheets and nanotubes which were recently summarized by us in this review article: (Li et al., *Chem. Rev.* 2021, 121, 14031-14087. DOI: 10.1021/acs.chemrev.1c00024). The first paper (Mannige et al. *Nature* 2015) reported a nanosheet structure, not a nanohelix structure. While the helical secondary structure of peptoid was mentioned in that study, that is very different from the supramolecular self-assembly structure. The “helical secondary structure” parallels similar concepts in peptides like α -helix and β -sheet. Therefore, our study is very different from that specific work.

In the second paper (Sun et al. *PNAS* 2020), we don’t see any mentions of “nanohelix” in that study at all, the authors did not report any nanohelix structures. Therefore, we do not agree with this reviewer that our study shares the concept of similar nanohelix formation.

In the third paper (Ghosh et al. Chem. Eur. J. 2020), Galia Maayan's group reported an interesting study of using metal-peptoid coordination to drive the formation of 3D single crystals. They further discovered that the aqueous solution of the re-dissolved 3D crystals contained helical fiber-like structures which the single crystal structure indicated the formation of coiled helical architecture as a result of self-assembly of the metallopeptoid duplexes. This study from Galia Maayan's group is very different from our work which was driven by non-covalent hydrophobic/hydrophilic interactions for the formation of nanohelices with controllable supramolecular chirality. Therefore, we do not agree with this reviewer for this statement "*The concept of similar nanohelix formation has been extensively reported by Zuckermann previously*" and believe the peptoid nanohelices with controllable supramolecular chirality reported here are highly novel and very different from various peptoid assemblies reported in the literature and mostly summarized in our recently reported reviewer article: Li et al., Chem. Rev. 2021, 121, 14031.

To address this comment and recognize the importance of these suggested papers in the area of peptoid assembly research, we now cited them as references 24, 25, and 26 in the revised manuscript:

interactions can be tuned by controlling the side chain chemistry^{22,23}. Recently, we and others discovered a series of amphiphilic peptoids that can assemble into diverse nanostructures,^{11,22-26} including membrane-mimetic nanosheets^{27,28} and nanotubes^{8,29} which. These structures share a similar molecular packing of amphiphilic peptoids akin to the lipid bilayer but these peptoid nanostructures are highly crystalline and robust. Despite the recent progress made in design and

2) "*Without a definite x-ray structure the claim of peptoid orientation can't be confirmed just using MD simulation.*"

Response: We agree that further characterizations could be helpful to determine the specific molecular packing model in the self-assembly. However, based on the previous work on amphiphilic peptoid self-assembly (Murnel et al. J. Am. Chem. Soc. 2010; Mannige et al. Nature 2015; Jin et al. Nat. Comm. 2016; Jin et al. Nat. Comm 2018;) and compare our XRD result with previous ones, we are confident to claim that our nanohelices follow the similar amphiphilic bilayer packing as the previous papers.

Besides, we hope this reviewer would appreciate the existing challenges of obtaining large enough crystals suitable for single crystal X-ray crystallography for this type of assembly peptoid system. That is also the main reason that we turned out helps to the solid-state NMR technique and MD simulations. Nonetheless, with our knowledge built from previous peptoid assembly system and all characterizations presented in this manuscript for the assembly of these short amphiphilic peptoids, we are able to confirm the peptoid orientation (i.e., bilayer-like packing of peptoids) within peptoid nanohelices.

3) "*Why only peptoid tetramer was used, a justification is missing. Definite control experiments without using Npm are lacking.*"

Response: We thank this reviewer for pointing out the importance of some control experiments. To further demonstrate the importance of hydrophobic interactions among Npm groups in the formation of nanohelices, we synthesized two additional peptoids Npm6Dig and Npm2Dig and investigated their self-assembly. Our results showed Npm6Dig with six Npm groups formed nanosheets while Npm2Dig with two Npm groups formed isolated particles, suggesting a medium hydrophobic interaction existed among Npm4 groups is important for the formation of peptoid nanohelices. Detailed description of these results was shown in our response to reviewer #1, question No. 2.

To address this comment, we added these following sentences highlighted in yellow:

“... a cis-conformation peptoid. To further address how hydrophobic interactions influence the peptoid assembly morphology, we varied the numbers of Npm groups from four to six and two. AFM results showed that the peptoid Npm6Dig with six Npm groups self-assembled into nanosheets (Supplementary Fig. 10a) due to enhanced hydrophobic interactions. In contrast, peptoid Npm2Dig with only two Npm groups self-assembled into isolated particles (Supplementary Fig. 10b) due to reduced hydrophobic interactions. Such enhanced hydrophobic interactions among Npm6 domains were also revealed in the XRD data of nanosheets assembled from Npm6Nce6 (Supplementary Fig. 11) in which two sharp peaks between d spacings of 2.9 Å and 4.6 Å were observed...”

4) *“Regarding the experiment with L-Ala and D-Ala is too primary. A few more amino acids especially with Phe and Lys in both L/D forms are required to comment on the reasons for the altering helices.”*

Response: We thank this reviewer for the interest in the amino-acid triggered change of supramolecular chirality of peptoid nanohelices. While we agree with this reviewer that more experiments could be done in this direction, we believe that our current results are enough to demonstrate: 1) the formation of these peptoid nanohelices is governed by the interplay of the hydrophilic and hydrophobic domains with the aqueous solvent; a slight change in the polar domain while keeping the hydrophobic Npm4 domain unchanged could dramatically influence the driving force that leads to the formation of peptoid helices. 2) The supramolecular chirality of assembled peptoid helices could be controlled by modifying assembling peptoids with a single chiral amino acid side chain. Thus, the additional modification of Npm4 domains with other amino acids and exploring the reasons behind the amino-acid-triggered change of nanohelix supramolecular chirality using computational simulations and other tools is beyond the scope of this paper. We hope this reviewer would agree with that.

To address this comment, we made these changes in the revised manuscript:

“...In future work, we would like to employ enhanced sampling molecular dynamics simulations and free energy calculations to gain molecular-level insight into the relationship between the peptoid side chain stereochemistry and the emergent supramolecular chirality and endeavor to rationalize the root of the opposite trend to that expected for peptide systems. Therefore ...”

5) *“Because our previous studies showed that the peaks that occurred in this area are corresponding to the presence of extensive π -stacking among aromatic side chain groups, the XRD results suggest that there are no π -stacking interactions among Npm side chains within Npm4Dig nanohelices.’ (Line 146-148) the reason why no π -stacking interactions have been observed is required.”*

Response: We thank this reviewer for pointing out this discussion about π -stacking interactions. While the exact reasons that led to the disappearance of those two XRD peaks are unknown, as shown in the Supplementary Figure 11, peptoids with six Npm groups (i.e., Npm6Dig and Npm6Nce6) formed a nice nanosheet structure (See newly added Figure in response to Reviewer #1, question No. 2), while peptoid with two Npm group (i.e., Npm2Dig) formed isolated particle. We hypothesize that the smaller number of Npm groups leads to the formation of more flexible bilayer packing of Npm4 groups which is hard to form a well-aligned π -stacking among aromatic side chain groups.

To address this comment, we made this following change (highlighted in yellow):

“In contrast to the XRD results of nanosheets assembled from peptoids with six Npm6 groups which have two significant peaks at 4.0 and 3.6 Å (Supplementary Fig. 11), no obvious peaks were observed in this area for self-assembled Npm4Dig nanohelices. Because our previous studies showed that the peaks occurred in this area are corresponding to the presence of extensive π -stacking among aromatic side chain groups, the XRD results suggest that there are no π -stacking interactions among Npm side chains within Npm4Dig nanohelices which could be due to the less crystallinity feature of these hydrophobic domains within nanohelices compared to those within crystalline nanosheets.”

6) “We thus replaced the achiral Dig group by L- or D-alanine group as the hydrophilic terminal of the peptoid molecules’ (Line 291-292), is it a typo that ala would be hydrophilic terminal as Ala is known to be hydrophobic?”

Response: Thanks for pointing this out, to address this comment and prevent confusion, we felt using the phase ‘polar domain’ would be a better description of this domain for these short amphiphilic peptoids. We have made corresponding changes in the revised manuscript.

7) “ ‘Supramolecular chirality’ is not supported with enough experiments, especially concentration-dependent CD is highly recommended.”

Response: We agree with the reviewer that CD is a powerful tool to confirm supramolecular chirality. However, for the assembled chiral nanomaterials from achiral molecules, the assembled peptoid nanohelices will need to be well-dispersed in solution in order to collect a decent circular dichroism (CD) spectroscopy. Due to the poor dispersity of these assembled peptoid nanohelices in water and the lack of CD signals for non-chiral peptoids, thus we are not sure if the concentration-dependent CD would give us much more useful information than the existed AFM and TEM images showing these chiral nanohelices. In contrast, to address this comment, we further did more SEM characterizations of these chiral nanohelices, and add them as the supplementary Figure 20. Therefore, with all AFM, TEM, and SEM images we presented in the revised manuscript, we believe we have enough experimental results to support the ‘supramolecular chirality’, hope this reviewer agree with that.

Supplementary Figure 20 | SEM images of the right-handed nanohelices self-assembled from Npm4-L-Ala at pH = 4.

We made these following changes in the revised manuscript:

both Npm4-D-Ala and Npm4-L-Ala formed gel-like materials. ADF-STEM, ~~and~~ AFM, ~~and~~ scanning electron microscopy (SEM) images confirmed that these gel-like materials contained a large amount of peptoid nanohelices with a dominant one single handedness (Fig. 4, ~~Supplementary~~ Figs. ~~16-20~~~~S11, S12, S13 and S14~~), which Npm4-D-Ala led to the formation of left-handed nanohelices while L-Ala led to the formation of nanohelices with all right-handedness.

8) ESI-MS ranges can be shown from 200-900 Da range for Fig S3. For Fig. S1, can you explain what is the peak for 459?

Response: To address this comment, we now used another set of UPLC-MS data of Npm4-D-Ala as updated Supplementary Figure 6.

Supplementary Figure 6 | UPLC-MS data of HPLC-purified peptoid: Npm4-D-Ala; the insert is the chemical structure of this peptoid.

The peak at 459 is corresponding to the total mass of (Npm3) shown below. It is the fragment of this whole peptoid during the MS measurement which we found that our UPLC-MS could function like MS-MS

technique when the gain voltage was high.

9) *“Formation of peptoid scaffold computationally could not be supported unless the lowest energy conformational search is performed.”*

Response: In the absence of any experimental results, we would agree with the reviewer’s assertion that the structure predicted by our unbiased simulations may or may not correspond to the equilibrium configurational state of the system and that enhanced sampling calculations designed to accelerate convergence to the free energy minimum would be warranted. Indeed, a large fraction of the work conducted in our group precisely employs this strategy. **In the present work, however, we combined our computer simulations with experimental analyses and observed an excellent agreement between the two.** Indeed, at the qualitative level the simulations and experiments and theory all identify a twisted ribbon morphology (Figs. 1-3) and we even see quantitative agreement between the pitch of the twist predicted by simulation and observed in experiment (“...the experimentally observed helical pitch of 86.4 ± 6.7 nm reported pH=7 in Fig. 3c is in excellent agreement with the pitch of ~ 87 nm predicted in our MD simulations...”). This level of agreement provides between simulation and experiment provides strong support that our molecular dynamics simulations are reliable and relax down into the thermodynamically preferred configurational state (i.e., the twisted ribbon) from the initial state (i.e., the planar ribbon) without the need for enhanced sampling techniques.

10) *“As a force field why CHARMM 22 is used could be explained in SI, comparing at least one geometry with CHARMM36 or NAMD.”*

Response: As stated in the Supplementary Information, the force field we used to model the peptoids is “the improved all-atom CGenFF peptoid force field developed by Weiser and Santiso as a peptoid-tuned modification of the CHARMM22 peptide force field”. Weiser and Santiso elected to use the CHARMM22 peptide force field as the basis for this peptoid force field and, as such, the CHARMM22 parameters are “baked into” the CGenFF peptoid force field. We concurred with the reviewer that it would be desirable to update the CGenFF peptoid force field on the basis of the newer CHARMM36 force field, but this is not a simple plug-and-play modification, but would rather require a complete reparameterization and, equally importantly, independent validation of the resulting model. Force field parameterization and validation is a computationally burdensome task that would require the investment of substantial computational effort and would likely result in a stand-alone publication, which is far beyond the scope of this study. As such, for the purposes of this work, we employed CGenFF as a modern and well-regarded peptoid force field with which we had prior experience and success (<https://doi.org/10.1021/acs.biomac.3c00107>). We also observed that this CGenFF force field has demonstrated quantitative accuracy in predicting the twist angle of the twisted ribbons in the present work and the diameter of self-assembled peptoid nanotubes in our prior work (<https://doi.org/10.1021/acs.biomac.3c00107>) leading strong support to its accuracy and reliability in modeling peptoid systems.

11) *“The referencing is not at all updated for the peptoids; crucial contributions in the field by others have not been acknowledged.”*

Response: Again, this reviewer probably had some misunderstandings or confusions about peptoid secondary structure vs the structure of peptoid assemblies. **In this paper, we are referring to the helical structure of assembled peptoids,** thus we do not feel that we have missed some crucial contributions in this area of self-assembled peptoid nanohelices. Nonetheless, given the importance of these suggested papers in the area of peptoid assembly research, we now cited them as references 24, 25, and 26 in the revised manuscript.

12) "I wonder where and how this tetramer peptoid can be used for 'generating a wide range of protein-like, functional nanomaterials as drug delivery agents, artificial enzymes, and tissue engineering scaffolds'? How can this tetramer contribute to drug delivery? Or tissue engineering? A few possible references or any primary experiment for the same would be nice to see as an application.

The conclusion part is poorly written, especially the last portion is misleading without proper experimental support."

Response: We thank the reviewer's suggestions about the conclusion part. To address this comment along with Review #1's comment, we made the following changes in the revised manuscript:

precisely control the structural organization of sequence-defined peptoids and their supramolecular chirality of their assemblies opens the door to a unique system suitable to reveal how molecular interactions between fundamental chemical moieties control the arrangement, packing, and folding assembly of biomacromolecules and inorganic particles,^{1,11,13,15-20,22} generating a wide range of protein-like, functional nanomaterials as nanocarriers for drug delivery agents,^{35,44,53} as artificial enzymes for catalytic reactions,^{8,54} and as tissue engineering scaffolds for biomimetic mineralization.^{43,55,56}

REVIEWERS' COMMENTS

Reviewer #1 (Remarks to the Author):

All the questions and concerns are clearly addressed. Although reviewer 3 pointed out the novelty in the first round of the review, the reviewer thinks that underlying science and potential impact is good enough for the publication in the prestigious journal such as Nature comm.

Reviewer #2 (Remarks to the Author):

The manuscript authors have addressed all of my concerns in their very thorough response to the reviewers. Their efforts have improved the manuscript.

Reviewer #3 (Remarks to the Author):

I am mostly convinced, though not sure about question 4 which was to explore the effect of other amino acids.

I thought the authors would perform two more syntheses (control experiments) attaching one hydrophilic and another hydrophobic amino acid with two separate peptoid tetramer (Npm4) strands to explore the role of amino acid only.

Reviewer #3 (Remarks to the Author):

"I am mostly convinced, though not sure about question 4 which was to explore the effect of other amino acids.

I thought the authors would perform two more syntheses (control experiments) attaching one hydrophilic and another hydrophobic amino acid with two separate peptoid tetramer (Npm4) strands to explore the role of amino acid only."

Response: We thank this reviewer again for the interest in the amino-acid triggered change of supramolecular chirality of peptoid nanohelices. While the additional modification of hydrophobic Npm4 domains with other amino acids is beyond the scope of this study, our recent preliminary result showed that peptoid Npm4-L-Asp which L-aspartic acid was used to modify the Npm4 group assembled into similar nanohelices to those assembled from Npm4-L-Ala.

[REDACTED]

To better address this comment further, we made these changes in the revised manuscript:

Supplementary Fig. 23c, d). In future work, we would like to employ enhanced sampling molecular dynamics simulations and free energy calculations to gain molecular-level insight into the relationship between the peptoid side chain stereochemistry and the emergent supramolecular chirality and endeavor to rationalize the root of the opposite trend to that expected for peptide systems. Our future direction also includes the attachment of the hydrophobic Npm4 domain with other single amino acids, such as hydrophilic aspartic acid with either L- or D-aspartic form, to investigate the influence of single chiral amino acid on the supramolecular chirality of peptoid nanohelices. Therefore, we ~~propose~~ reason that ~~the modification~~ modifying the hydrophobic domain of short assembling peptoids with single amino acid sequences could provide a new platform for controlling the formation of helical nanostructures showing dramatic differences in supramolecular chirality control from those assembled from peptides and proteins.

Discussion